# Anatomy and Comparative Transcriptome Reveal the Mechanism of Male Sterility in *Salvia miltiorrhiza*

**DOI:** 10.3390/ijms241210259

**Published:** 2023-06-17

**Authors:** Jinqiu Liao, Zhizhou Zhang, Yukun Shang, Yuanyuan Jiang, Zixuan Su, Xuexue Deng, Xiang Pu, Ruiwu Yang, Li Zhang

**Affiliations:** 1College of Life Science, Sichuan Agricultural University, Ya’an 625014, China; 2Sichuan Provincial Engineering Research Center for Breeding Technology of Authentic Traditional Chinese Medicine, Sichuan Agricultural University, Ya’an 625014, China; 3College of Science, Sichuan Agricultural University, Ya’an 625014, China

**Keywords:** pollen wall, tapetum, starch grain, transcriptomics, male sterility

## Abstract

*Salvia miltiorrhiza* Bunge is an important traditional herb. *Salvia miltiorrhiza* is distributed in the Sichuan province of China (here called SC). Under natural conditions, it does not bear seeds and its sterility mechanism is still unclear. Through artificial cross, there was defective pistil and partial pollen abortion in these plants. Electron microscopy results showed that the defective pollen wall was caused by delayed degradation of the tapetum. Due to the lack of starch and organelle, the abortive pollen grains showed shrinkage. RNA-seq was performed to explore the molecular mechanisms of pollen abortion. KEGG enrichment analysis suggested that the pathways of phytohormone, starch, lipid, pectin, and phenylpropanoid affected the fertility of *S. miltiorrhiza*. Moreover, some differentially expressed genes involved in starch synthesis and plant hormone signaling were identified. These results contribute to the molecular mechanism of pollen sterility and provide a more theoretical foundation for molecular-assisted breeding.

## 1. Introduction

In plants, male sterility affects reproduction seriously, and it is caused by ribosome defects, early or delayed degradation of the tapetum, the defective pollen wall, and abnormal anther differentiation [1,2,3]. Therefore, lots of genes are involved in regulating pollen development and it is a very complex process [4,5,6].

Pollen development consists of two phases: microsporogenesis and male gametogenesis [7]. The sporogenous cells in the pollen sac form a large number of microspore mother cells through mitosis. The microspore mother cell is diploid and has two successive meiotic divisions, resulting in the microspore tetrad. Each microspore has a thick callose wall and a thin pectin wall [8]. Through tapetum degradation, sugars, proteins, amino acids, and sporopollenin precursors are released to promote the development of pollen grains [9,10]. Meanwhile, there are significant changes in the composition and structure of the pollen wall [11]. Finally, the pollen grains need to accumulate energy for germination [12,13].

Timely degradation of tapetum cells is vital for pollen development and its disturbance usually causes male sterility. Besides nutritional functions, tapetal degradation also promotes the degradation of callose and provides materials for pollen wall formation [14,15]. In *Arabidopsis*, it is reported that a regulatory network: “DYSFUNCTIONAL TAPETUM 1 (*DYT1*), Defective in Tapetal Development and Function 1 (*TDF1*)-ABORTED MICROSPORES (*AMS*)- MALE STERILITY 88 (*MS188*)-MALE STERILITY 1 (*MS1*)” participates in regulating tapetal degradation and pollen fertility [16,17,18]. For example, *TDF1* expression is induced by *DYT1* in the tapetal, and both of its mutants exhibit vacuolated microspores and a defective pollen wall [18,19]. Therefore, the identification of genes regulating tapetal degradation can contribute to a deep understanding of the molecular mechanism of male sterility.

*Salvia miltiorrhiza* Bunge, an important traditional Chinese herb, has a significant effect on cardiovascular disease treatment with great market demand [20,21,22]. *Salvia miltiorrhiza* is distributed in the Sichuan province of China (here called SC). For a long time, our group finds that the individuals of the SC population do not produce seeds under natural conditions [23]. Although the SC population is known for its high content of salvianolic acid B, its yield and quality are severely reduced with long-term asexual reproduction. Thus, the analysis of the origin of *S. miltiorrhiza* sterility will help people to cultivate male sterile lines, and provide germplasm resources for artificial crossbreeding in the future. A previous study analyzed physiological characteristics and metabolic differences in leaves of the sterile line of *S. miltiorrhiza* [24]. In the hybrid offspring of *S. miltiorrhiza* and *S. miltiorrhiza* f. *alba*, male sterility was caused by abnormal microgametogenesis, and key genes were initially screened through a comparative transcriptome analysis [25]. However, there are still many potential molecular mechanisms related to the sterility of *S. miltiorrhiza* that are worth further research. In addition, *S. miltiorrhiza* is also considered an important medicinal model plant with a small genome size, short life cycle, etc [26]. Therefore, this sterile material of *S. miltiorrhiza* can reveal a new molecular mechanism and lay a theoretical foundation for other medicinal plants in the future.

In this study, morphology and electron microscope technology were performed to characterize the differences between fertile and sterile pollen grains of *S. miltiorrhiza*. Through comparative transcriptome analysis at different stages, we preliminarily screened some potential pathways and candidate genes for male sterility. These findings can provide new insights into the molecular mechanism of male sterility in *S. miltiorrhiza.*

## 2. Results

### 2.1. Defective Pistil and Pollen Grains Jointly Cause Sterility in SC

To observe the differences between SD (S. miltiorrhiza distributed in the Shandong province of China) and SC, we examined phenotypic and pollen-setting rates. Compared with SD, the inflorescence axis of SC was obviously shorter (Figure 1A). In addition, the flower buds and anthers in SC were significantly smaller than SD (Figure 1B,C). Whether in the natural state or under artificial bagging conditions, SC had no seeds per floret and its seed-setting rate was significantly lower than SD (Figure 1E,F).

To further understand the cause of infertility in SC, artificial pollination experiments were conducted. We pollinated SD with the mature pollen grains of SD and SC. These offsprings were named SDZJ and SDSC, respectively. When the stigmas of SC were crossed with SD and SC pollens, these offsprings were named SCSD and SCZJ, respectively. In Figure 1G, the setting rates of SDSC and SCZJ were significantly lower than the setting rates of SDZJ and SCSD, respectively. Interestingly, the setting rate of SDZJ was significantly higher than SCSD, and a similar phenomenon was observed between SDSC and SCZJ. These results of artificial pollination showed that the defective pistil and pollen grains jointly cause sterility in SC. Next, we mainly focused on male sterility in *S. miltiorrhiza*. In the future, we will carry out other research related to the defective pistil.

In order to clarify the sterility of *S. miltiorrhiza* caused by pollen abortion, the pollen vitality was tested (Figure 1D). With the anther’s normal dehiscing, SD released a lot of mature pollen grains; however, SC had fewer pollen grains and its pollen vitality was lower than SD (Figure 1H). It proved that the abortion of pollen grains caused male sterility in SC.

### 2.2. Sunken Pollen Grains and Impaired Pollen Walls Lead to Partial Abortion of SC Pollen Grains

To further observe the differences between their pollen grains, we performed scanning electron microscope analysis (SEM). In Figure 2A,B, SD pollen grains were obvious oval lumina with smooth pollen wall and germinal furrows. SD pollen grains had finely reticulated ornamentation on their surfaces (Figure 2C). In SC, there were still a few plump round pollen grains, which were consistent with the fertile pollen grains in SD (Figure 2D,E). It was speculated that they were fertile pollen grains and the wrinkled pollen grains in SC were sterile pollen grains (Figure 2E). Subsequently, fertile and sterile pollen grains were selected for further observation of the differences in the pollen wall. In comparison with fertile pollen grains, the sterile pollen wall was cohesive (Figure 2C,F). These results showed that the collapsed pollen grains and the defective pollen wall caused pollen abortion in SC.

### 2.3. Defective Pollen Cell Walls in SC Caused by Delayed Degradation of Tapetum

To further understand the cytological causes of pollen wall malformations, the different periods of anther development were observed (Figure 3). At the pollen mother cell stage (PMC), there was no obvious difference in the structures of microsporangia between SD and SC (Figure 3A,B). At the meiosis stage (MEI), the tapetum in SC was significantly swollen and vacuole (Figure 3C,D). During the tetrad period stage (TD), the tapetum in SD began to break away from the endothecium and disintegrate gradually, while this process has been relatively delayed in SC (Figure 3E,F). In the period of the young microspore stage (YM), the tapetum cells completely disappeared in SD, and the band-like thickening occurred on the tangential and transverse cell wall to form a fiber layer (Figure 3G). However, there were still lots of tapetum cells in SC and the cell wall inside had not been thickened (Figure 3H). Starch was stored in fertile pollen grains at the final stage (mature pollen stage, MP) to provide energy for pollen maturation and germination (Figure 3I). In contrast, there were many wrinkled immature pollen grains that had not been successfully stained in SC (Figure 3J).

### 2.4. Defects of the Microsporangium Wall and the Lack of Starch Lead to the Pollen Sterility

Subsequently, we performed transmission electron microscopy (TEM) analysis to obtain more details between SD and SC (Figure 4). Ubisch bodies were found on the surface of the SD tapetal at the TD stage; however, a similar phenomenon was not observed in SC (Figure 4C,D). In Figure 4E, some thin structures were observed clearly on the surface of tetrads in fertile pollen. The tapetal was in the process of degradation and there were a lot of Ubisch bodies on their surfaces (Figure 4G,I). At the YM stage, the microspores formed dense cytoplasm with the tapetal degradation in SD. In contrast, the microspore appeared the vacuolization with a massive cytoplasmic loss in SC. The tapetal cell wall was obvious, which indicated that tapetal cells had not yet disintegrated, and the number of Ubisch bodies on it was small (Figure 4H,J). At the YM stage, microspores were forming the exine, tectum, and baculum (Figure 4K,L). At the MP stage, there were a lot of organelles and starch grains in the fertile pollen grains (Figure 4M,N). In contrast, vacuolization appeared in the sterile pollen grains with lacking starch grains and organelles (Figure 4O). The tectum of fertile pollen grains was connected with the nexine by the baculum, which was the basis for the formation of the exine (Figure 4P). Due to the absence of the baculum, the tectum of the sterile pollen collapsed and the exine structure was disordered (Figure 4Q).

### 2.5. Identification of Differentially Expressed Genes (DEGs) between SD and SC at the Different Stages

To reveal the molecular mechanism of male sterility in SC, a comparative transcriptome analysis was performed using anthers of SD and SC at the three different developmental stages, i.e., MEI, YM, and MP. After data cleaning and quality control, in each library, we generated 39,864,920–47,676,270 clean reads mapped to the whole genome to obtain positional information and sequence information specific to the sample. The match rate was in the range of 86.02−90.64%, indicating that the data is available for subsequent analysis. Principal component analysis showed good repeatability and strong correlation among different groups (Figure 5A). We randomly selected nine genes to validate the transcriptomic results by qRT-PCR, which confirmed the reliability of the transcriptomic data (Appendix A). In Figure 5B, 97 DEGs showed significant differences in the combinations of SCMEI vs. SCYM, SDMEI vs. SDYM, SCYM vs. SCMP, and SDYM vs. SDMP. Subsequently, we identified 36 transcription factors (TFs) from 97 DEGs, and these TFs are divided into 25 TF families (Figure 5C). There are some TFs belonging to Pkinase (8.05%), MYB (8.05%), GRAS (2.72%), Cu_bind_like (11.17%), and WRKY (2.72%) families.

To identify the essential genes regulating tapetal degradation and microspore development, ninety-seven DEGs were divided into eight subclusters based on their accumulation patterns (Figure 5D,E, and Appendix A). Based on our data, the expression levels of these genes (subcluster_3 and subcluster_5) reached a peak at the YM stage and subsequently decreased. Except for the MEI stage, genes were more expressed in SC, suggesting that subcluster_3 might be related to male sterility. Among these DEGs, SmiChr064206, encoding a WRKY6 TF, caught our attention. Additionally, previous studies showed that WRKY6 participated in pollen maturation and regulated tapetal degradation [27,28]. Compared with subcluster_3, DEGs in subcluster_4 showed a completely opposite expression pattern and the expression level increased significantly in the MP stage. For example, SmiChr082452 encoded a polygalacturonase, which might be involved in regulating the pollen wall formation.

With the pollen development, the expression of these DEGs continuously decreased (subcluster_2 and subcluster_7) and continuously increased (subcluster_1, subcluster_6, and subcluster_8). Particularly, DEGs in subcluster_7 and subcluster_8 had lower expression in SC than SD at YM and MP stages. SmiChr052849 (stamen-specific protein, FIL1) was found in cluster 7, indicating that its downregulation might cause pollen sterility in SC. For example, SmiChr013751 and SmiChr022309 were involved in phenylpropanoid biosynthesis and pectate biosynthesis, respectively, which might be associated with the formation of the pollen wall. The expression level of SmiChr061226 (an anther-specific protein) was lower in SC at the YM stage, which might cause the pollen abortion in SC.

### 2.6. Transcriptome Analysis Revealed Potential Pathways Related to Male Sterility

In order to recognize the differences between SD and SC at the different stages, we conducted an enrichment analysis on their different stages. From the MEI stage to the YM stage, gene ontology (GO) enrichment analysis showed that a few biological processes (BP) related to lipids were only enriched in SCMEI vs. SCYM, such as lipid biosynthetic process (GO: 0008610) and lipid metabolic process (GO: 0006629), comparing with SDMEI vs. SDYM (Figure 6A,C). Hence, we speculated that this might cause the difference between SD and SC in the composition of pollen exine. In the “cell component” (CC), lots of DEGs mainly participated in the integral component of membrane (GO: 0016021), intrinsic component of membrane (GO: 0031224), membrane (GO: 0016020), and membrane part (GO: 0044425). From the YM stage to the MP stage, we also found that cell wall modification (GO: 0042545), cell wall organization (GO: 0071555), cell wall (GO: 0005618), and pectinesterase activity (GO: 0030599) were highly enriched (Figure 6B,D). These results indicated that there were a few differences between SD and SC in the formation of the pollen wall, which was consistent with TEM analysis (Figure 4M–Q).

Through tapetal degradation, sporopollenin was released to form the exine of the microspore [29]. According to the KEGG enrichment analysis, phenylpropanoid biosynthesis (ko: 00940), flavonoid biosynthesis (ko: 00941), flavone and flavonol biosynthesis (ko: 00944), starch and sucrose metabolism (ko: 00500), pentose and glucuronate interconversions (ko: 00040), and other pathways were significantly enriched at the early anther development (Figure 6E,G). Compared with SDYM vs. SDMP, starch and sucrose metabolism (ko: 00500) and plant hormone signal transduction (ko: 04075) were still significantly enriched in SCYM vs. SCMP (Figure 6F,H). In the plant hormone signaling pathway, we found that most DEGs related to IAA (41/71) and ABA (13/71) showed increased expression at the MP stage (Appendix A and Appendix A). Simultaneously, DEGs related to starch biosynthesis in SCYM vs. SCMP might cause the loss of starch granules in a sterile pollen grain (Appendix A and Appendix A). These results indicated that there were some special changes in the process of SC anther development, which led to its sterility.

## 3. Discussion

Sporopollenin consists of many complex compounds, such as flavonoids and fatty acids [30]. The sporopollenin is not transferred to microspores timely, resulting in aberrant exine [31]. When sporopollenin is deposited on the microspores, the structure of the exine and bacula gradually forms on the primexine [32,33]. Therefore, delayed degradation of the tapetum can affect the correct transport of sporopollenin precursors to the pollen wall, causing a defective pollen wall. In this study, the hexagonal pattern on exine is clearer in fertile pollen grains than sterile pollen grains, indicating the presence of enough matrix to accumulate sporopollenin (Figure 2C,F). TEM analysis also supports this conclusion (Figure 4P,Q). As a signal molecule, ROS plays a key role in regulating tapetal degradation [34,35]. The downregulation of antioxidant enzyme genes affects tapetal degradation and causes vacuolization of microspores, resulting in pollen abortion [36]. In subcluster_7, we also notice an encoding peroxidase gene SmiChr013751, and it shows a lower expression level in SC than SD during tapetal degradation (Figure 5E). A previous study showed that an A9 (anther-specific protein) promoter was activated solely in tapetum cells to regulate the timely degradation of tapetum in tobacco [37,38]. In addition, the expression level of SmiChr061226 (an anther-specific protein) is lower than SD at the YM stage (Figure 5E). Thus, we speculate that these DEGs may lead to the difference in the pollen wall composition between SD and SC.

Starch is the main energy source and lack of starch in mature pollen grains causes male sterility [39]. Through TEM analysis, there are many starch grains in fertile pollen grains. Conversely, sterile pollen grains do not have any substantial filling and they are shriveled (Figure 2E and Figure 4M–O). As a precursor of starch synthesis, trehalose is important for regulating plant development and the downregulation of trehalose-phosphate phosphatase (TPP) proteins in anthers cause male sterility [40,41]. TPP (PAXXG016120) is expressed in mature pollen of *Phalaenopsis aphrodite* to accumulate high levels of sugars, and it provides nutrients and sucrose for pollen germination [41]. Similarly, we find that DEGs encoding TPP protein have higher expression levels in SD than SC at the same stage, such as SmiChr073137 (Appendix A). In rice, the absence of hexokinase5 (HXK5) reduces hexokinase activity and starch content, resulting in male sterility [42]. Among these DEGs, SmiChr063059 is annotated as hexokinase proteins and it is downregulated in SCMP than SDMP (Appendix A). Thus, downregulated DEGs related to starch synthesis may contribute to male sterility in *S. miltiorrhiza*.

Plant hormones regulate many biological processes, such as plant growth and anther development. Compared with SDYM vs. SDMP, we notice that plant hormone signal transduction (ko: 04075) is still significantly enriched in SC at the same stage (Figure 6F,H), and more than half of the DEGs were involved in the auxin (IAA) signal. At the late stage, IAA accumulates significantly to promote starch synthesis and pollen germination [43]. However, the over-expression of CASEIN KINASEI (*GhCKI*) inhibits starch synthase activity and promotes IAA accumulation in cotton, resulting in pollen sterility [44]. In addition, we find that the expression levels of most genes related to IAA signal are the highest at the MP stage. These results indicate that IAA may affect starch biosynthesis at the MP stage in *S. miltiorrhiza*.

Besides IAA, a lot of DEGs participate in the abscisic acid (ABA) signal pathway, and there is more evidence that ABA regulates pollen fertility by mediating sugar metabolism and transport [45,46]. The increase in ABA content inhibits the expression of *OsINV4* and *OsMST8*, and induces the upregulation of *OsMST7*. This leads to incomplete pollen wall and loss of starch in sterile pollen grains [45]. According to transcriptome analysis, the genes encoding SNF1-related protein kinases 2 (SnRK2) are activated in SC at the MP stage, such as SmiChr032403 (Appendix A). In plants, ABA, PYR/PYL/RCAR receptors, and type 2C protein phosphatases form complexes to release SnRK2s. These SnRK2s activate downstream components to modulate the ABA response. These potential clues suggest that SC may accumulate too much ABA to interfere with starch biosynthesis at the MP stage (Figure 4O). As a positive regulator in ABA signaling, *AtWRKY6* alters fatty acid compositions in *Arabidopsis* [27,28]. Moreover, ABA also controls tapetal degradation and the development of microspores [47]. In subcluster_3, SmiChr064206 named WRKY6 may affect tapetal degradation and ABA signal pathway (Appendix A).

## 4. Materials and Methods

### 4.1. Plant Materials and Growth Conditions

The fertile *S. miltiorrhiza* (SD) and the sterile *S. miltiorrhiza* (SC) were both planted in the field at Zhongjiang (Deyang, Sichuan province, China, 30°55′8.22″ N, 104°37′27.16″ E) experimental base of Sichuan Agricultural University. They each planted about 3000 plants with 0.4 m spacing and 0.5 m rows spacing.

### 4.2. Artificial Pollination

Artificial pollination was conducted in early June 2021. In September 2021, the number of seeds in four different treatments, i.e., seed setting rate under a natural state and artificial pollination (including selfing and reciprocal cross) was counted.

### 4.3. Phenotypic Analysis and Pollen Viability Analysis

Photographs of the verticillaster, corolla, and anther were obtained using a Canon 80D digital camera (Canon, Tokyo, Japan) during the flowering period.

After the anthers were dispersed, the mature pollen grains were collected from 11 individuals of SD and SC, respectively. Pollen grains were stained with 1% I_2_/KI solution [48]. We observed and counted at least 30 visual fields, respectively, under a Nikon ECLIPSE CI microscope (×200) (Nikon, Tokyo, Japan) for pollen vitality statistics, and then took photos.

### 4.4. SEM Analyses

The anthers (SD and SC) were washed twice with phosphate-buffered saline (PBS) for 5 min each time and dehydrated with a graded ethanol series (30%, 50%, 70%, 80%, 90%, 95%, 100%, and 10 min per grade). The sample was gently stuck to the conductive adhesive and sprayed with ion sputtering. Under the microscope (JEOL JSM-6360LV, Japan Electron Optics Laboratory, Tokyo, Japan), pollen grains on the endothecium in SD and SC were selected for observation at an accelerating voltage of 20 kV [49].

### 4.5. Paraffin Section Observation

In order to understand the development process of pollen grains, the anthers at the different stages of SD and SC were fixed in the formaldehyde-acetic acid-ethanol (FAA) and dehydrated in an ethanol series. The sizes of anthers at different stages were as follows: pollen mother cell stage (0.05–0.08 mm), meiosis stage (0.10–0.15 mm), tetrad period stage (0.20–0.22 mm), young microspore stage (0.25–0.33 mm), and mature pollen stage (0.40–0.50 mm). Anther sections were obtained using paraffin inclusion according to previous research [25]. The samples were embedded in paraffin (Hualing, Shanghai, China). The 5 µm-thick microtome sections were placed onto gelatin-coated glass slides (Solarbio, China) and stained with safranine and fast green double dyeing (Solarbio, Beijing, China). The sections of anthers at different stages were observed using a Nikon DS-Ri1-U3 Digital camera mounted on a Nikon ECLIPSE CI microscope.

### 4.6. TEM Analyses

According to the results of the paraffin section, the anthers at the tetrad (TD), the young microspore (YM), and the mature pollen (MP) stages were prefixed with 2.5% glutaraldehyde. Then, the tissue was post-fixed in 1% osmium tetroxide, dehydrated in a graded acetone series, infiltrated in Epox 812 for longer, and embedded. The semi-thin sections were stained with methylene blue and ultrathin sections were cut with a diamond knife into thin sections (70 nm), and then stained with 2% (*v*/*v*) aqueous uranyl acetate and lead citrate. Finally, the sections were examined using JEM-1400-FLASH transmission electron microscope at 80 kV (Japan Electron Optics Laboratory, Tokyo, Japan) [50].

### 4.7. Transcriptome Sequencing Analysis

After the total RNA of *S. miltiorrhiza* was isolated, the library was constructed, and the Illumina Novaseq platform (Novogene, Beijing, China, https://www.novogene.com (accessed on 29 September 2021)) was used for sequencing. The genes were filtered with DESeq2 with a |log2FC| > 2 and false discovery rate (FDR) <0.05 were defined as differentially expressed genes (DEGs). Gene ontology (GO) enrichment analysis and *Kyoto Encyclopedia of Gene and Genomes* (KEGG) enrichment analysis were corrected *p*-value < 0.05.

### 4.8. Q-PCR Assay/Quantitative Realtime-PCR (qRT-PCR) Validation

First-strand DNA was synthesized using a transcript Uni All-in-One First-Strand cDNA Synthesis SuperMix for qPCR (TransGen Biotech, Beijing, China) according to the manufacturer’s instructions. *SmHIATL1* and *SmYippee* genes were used as internal controls for the next experiment. The primers used for reverse transcription quantitative PCR were designed using Primer Premier 6, as listed in Appendix A. The specific primers for qRT-PCR were performed on the CFX-96 Real-Time PCR Detection System (Bio-Rad, Berkeley, CA USA) using PerfectStart Green qPCR SuperMix (TransGen Biotech, Beijing, China). Finally, the expression of the differential gene was calculated using formula 2^−ΔΔCt^ [51].

## 5. Conclusions

The low fertility of SC is caused by a defective pistil and pollen grains. Through the paraffin section, we find that delayed degradation of the tapetum is one of the reasons for male sterility. Integrated SEM and TEM analysis shows that the sterile pollen grains lack starch grains filling and have a defective pollen wall. According to comparative transcriptome analysis, we identify some potential pathways (phenylpropanoid, flavonoid, starch, pectin, phytohormone, etc.) which affect the biosynthesis of pollen walls and starch grains. Among them, SmiChr064206 and SmiChr073137 affects the delayed degradation of tapetum and impairs starch accumulation in SC, respectively, leading to male sterility. Taken together, our findings provide valuable clues and key candidate genes for further research on the male sterility of *S. miltiorrhiza*.

## Figures and Tables

**Figure 1 ijms-24-10259-f001:**
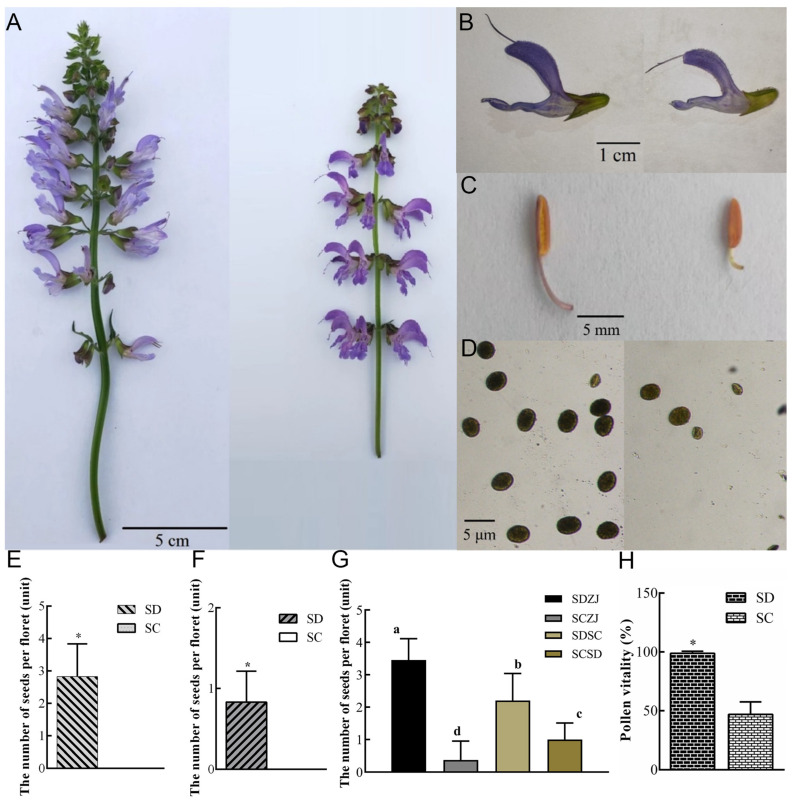
Phenotypic and pollen vitality rate. (**A**) the inflorescence of SD (**left**) and SC (**right**) at the flowering period; (**B**) the Corolla of SD (**left**) and SC (**right**); (**C**) the anther of SD (**left**) and SC (**right**); (**D**) the pollen grains of SD (**left**) and SC (**right**); (**E**) the number of seeds per floret in SD and SC which are planted in the natural state; (**F**) the number of seeds per floret in SD and SC under the condition of artificial bagging; (**G**) the number of seeds per floret in the artificial pollination offspring. (**H**) the proportion of pollen vitality of SD and SC in a natural state. SD and SC referred to S. miltiorrhiza distributed in Shandong province and Sichuan province of China, respectively. SDZJ and SDSC were the offspring produced by SD which was crossed with the mature pollen grains of SD and SC, respectively. SCSD and SCZJ were the offspring produced by SC which was crossed with the mature pollen grains of SD and SC, respectively; data were the means ± standard deviation (*n* = 20, 30), and data in E, F, and H analyzed by one-way ANOVA with LSD test for a comparison of significance. * indicated statistically significant differences at *p* < 0.05. Data in G was analyzed by Duncan’s new complex range method. a, b, c, d indicates statistically significant differences at *p* < 0.05.

**Figure 2 ijms-24-10259-f002:**
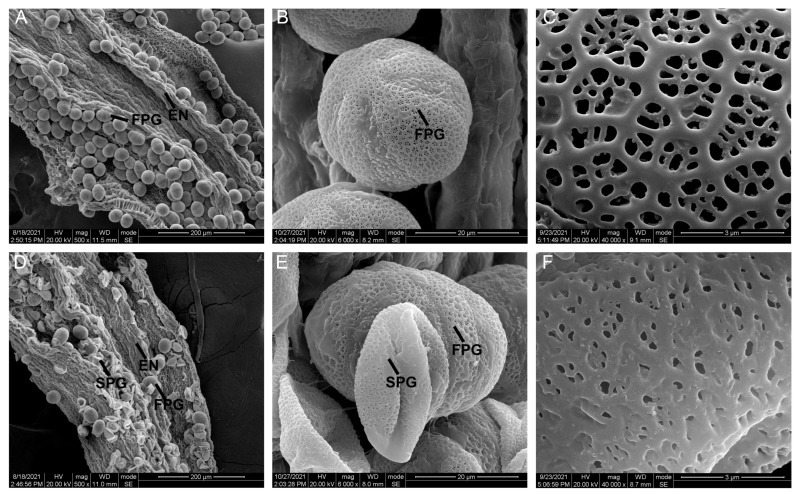
Scanning electron microscopy of cracking anthers and mature pollen grains. (**A**,**D**) the cracked anthers in SD and SC; (**B**) the mature fertile pollen grains in SD; (**C**) the surface of mature fertile pollen grains in SD; (**E**) the mature pollen grains in SC; (**F**) the surface of mature sterile pollen grains in SC. SD and SC referred to *S. miltiorrhiza* distributed in Shandong province and Sichuan province of China, respectively. EN, endothecium; FPG, fertile pollen grain; SPG, sterile pollen grain.

**Figure 3 ijms-24-10259-f003:**
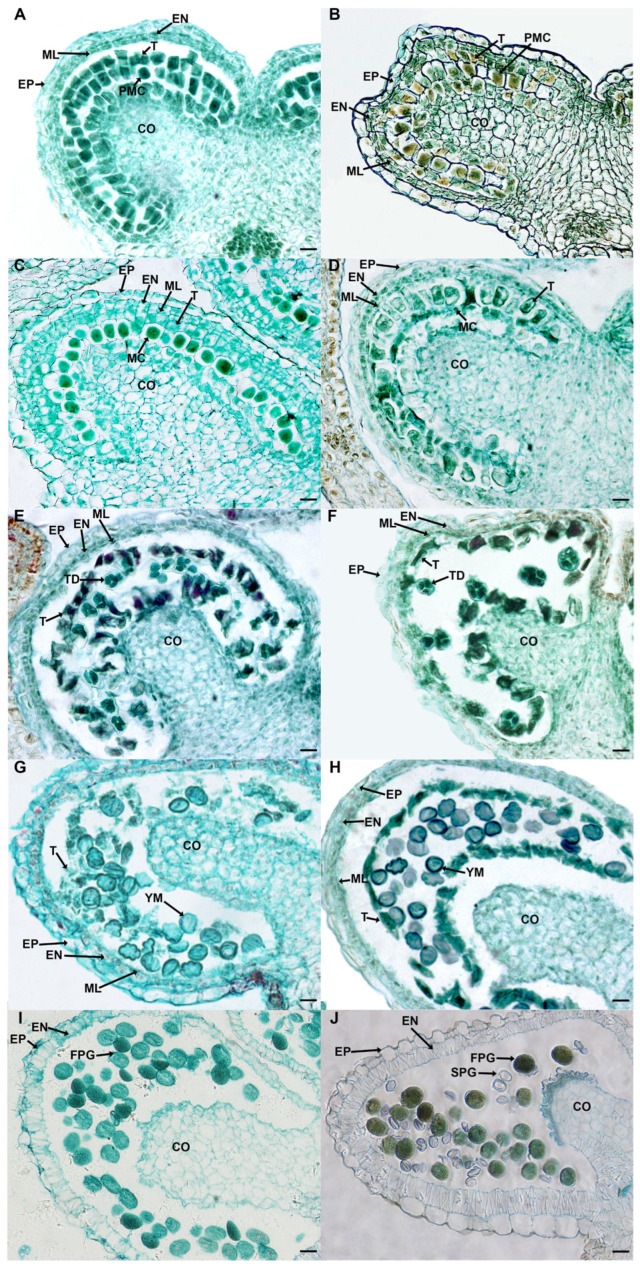
Comparison of anther development in SD and SC at the different stages by semithin sections. (**A**,**B**) the pollen mother cell stage (PMC) of SD and SC; (**C**,**D**) the meiosis stage (MEI) of SD and SC; (**E**,**F**) the tetrad (TD) stage of SD and SC; (**G**,**H**) the young microspore (YM) stage of SD and SC; (**I**,**J**) the mature pollen (MP) stage of SD and SC. SD and SC referred to *S. miltiorrhiza* distributed in Shandong province and Sichuan province of China, respectively. CO, connective; EN, endothecium; EP, epidermis; ML, middle layer; T, tapetum; FPG, fertile pollen grain; SPG, sterile pollen grain. bar = 200 μm.

**Figure 4 ijms-24-10259-f004:**
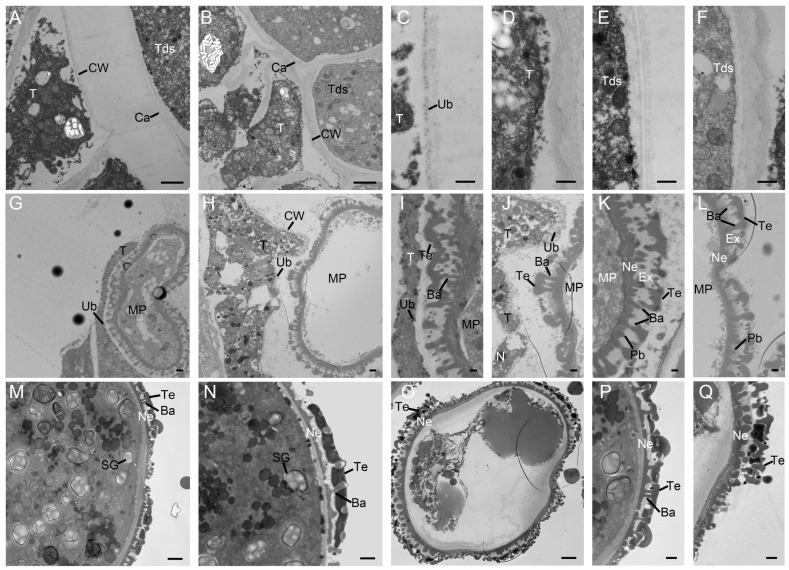
Transmission electron microscope (TEM) analysis of the anthers from the SD and SC. (**A**–**F**) the tetrad stage, tapetal and tetrad surfaces of SD (**A**,**C**,**E**) and SC (**B**,**D**,**F**); (**G**–**L**) the young microspore stage, tapetal and microspore of free microspores of SD (**G**,**I**,**K**) and SC (**H**,**J**,**L**). (**M**–**Q**) the mature pollen stage, pollen grains, and exines of SD (**M**,**P**) and SC (**N**,**O**,**Q**). SD and SC referred to S. miltiorrhiza distributed in Shandong province and Sichuan province of China, respectively. Ba, bacula; Ca, callose; CW, cell wall; En, Endothecium; Ex, exine; ML, middle layer; MP, microspore; N, nucleus; Ne, nexine; T, tapetum; Tds, tetrads; Ub, Ubisch body; Te, tectum; SG, starch granule. Bar = 2 μm (**A**, **B**, **G**, **H**, **M**, **N**, and **O**); bar = 0.5 μm (**C**–**F**); bar = 1 μm (**I**–**L**, **P**, and **Q**).

**Figure 5 ijms-24-10259-f005:**
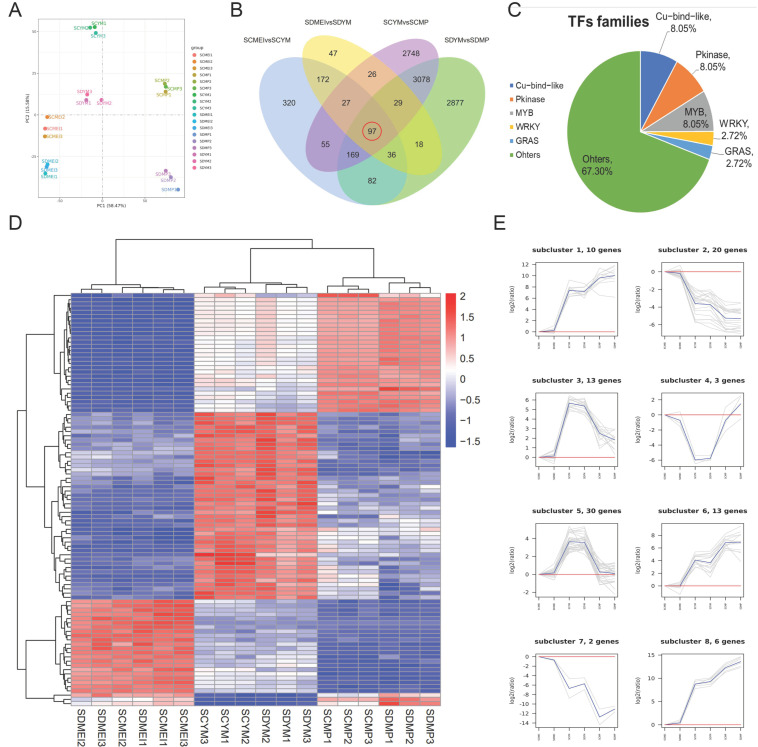
Preliminary analysis of transcriptomic data. (**A**) Principal Component Analysis (PCA); (**B**) Venn diagram of differentially expressed genes (DEGs); (**C**) distribution of differentially expressed TFs; (**D**) heatmap of 97 DEGs; (**E**) dynamics of 97 DEGs expression pattern during the different stages.

**Figure 6 ijms-24-10259-f006:**
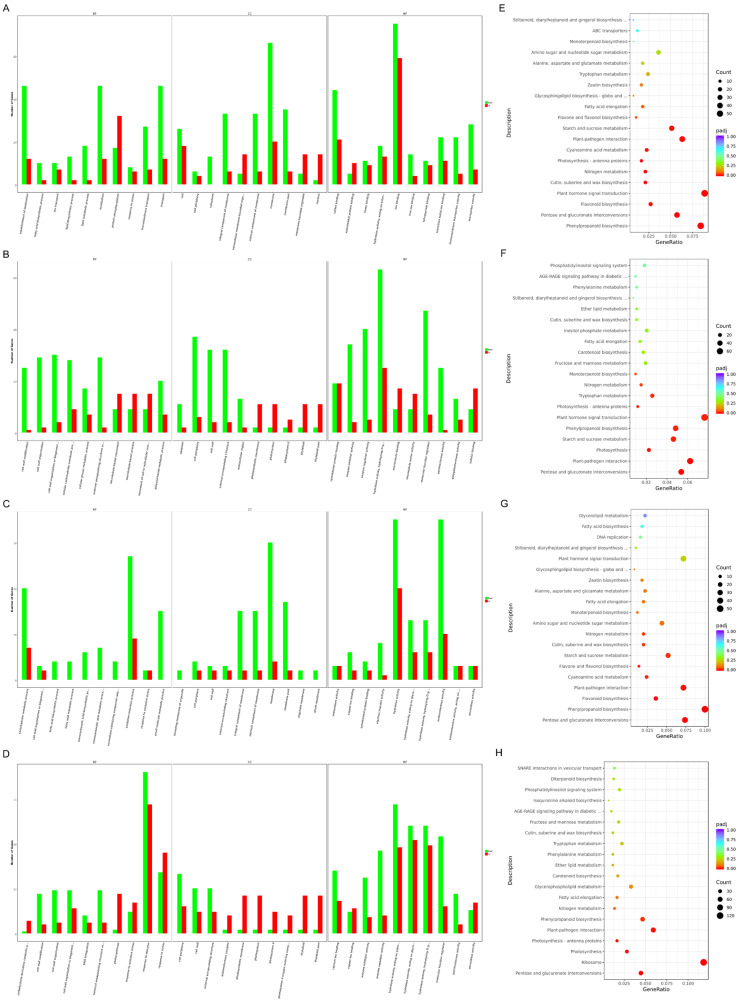
GO and KEGG enrichment analysis. (**A**–**D**) GO enrichment analysis from left to right: SCMEI vs. SCYM, SCYM vs. SCMP, SDMEI vs. SDYM, SDYM vs. SDMP; (**E**–**H**) top 20 KEGG pathways with the most significant enrichment from left to right: SCMEI vs. SCYM, SCYM vs. SCMP, SDMEI vs. SDYM, SDYM vs. SDMP.

## Data Availability

In this study, transcriptomics was deposited in the NCBI BioProject under accessions PRJNA863332.

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
