# Peer review of "Anatomy and Comparative Transcriptome Reveal the Mechanism of Male Sterility in Salvia miltiorrhiza"

_ijms, 2023, doi:10.3390/ijms241210259_

Round 1

Reviewer 1 Report

Readability needs some kind of improvement, with an English review.

See: For a long time, our group finds that S. miltiorrhiza distributed in China (Sichuan province, SC) does not produce seeds under natural conditions [23].

The text presents an abbreviation for a region: China (Sichuan province, SC). Nonetheless, this abbreviation is used in the present work for the plants, and not for a region. So, I recommend an unequivocal explanation of the SC abbreviation: Salvia miltiorrhiza is distributed in China, Sichuan province (here called SC plants). For a long time, our group finds that the individuals of the SC population do not produce seeds under natural conditions [23].

Several symbols and abbreviations are not explained in the figures and text. Graphs should be self-explanatory. Legends must contain all abbreviations and symbols used in each graph.

Both abbreviations SC and SD must be in the same order throughout the manuscript. Please use only one pattern in all text and captions for figures and tables.

SC first and SD after

or

SD first and SC after

It will improve readability.

A revision of this sequence must be done in all manuscript.

Several species are without italics and their taxonomic authorities are missing. Taxonomic authority is required for the first citation of a species. The species first citation is with the genus in full; after that, the genus is abbreviated, unless there is a risk of misunderstanding. You need to review the entire manuscript in view of these aspects.

The are no citations and references for some methodological procedures, such as pollen fertility, SEM and TEM analyses, and histological exam. Which is the literature used in these procedures? The methodology on pollen grain vitality is lacking. Please report this methodology or point to the literature on which this procedure is based.

Review "hybrid" terminology throughout the manuscript.

SDZJ = SD ovule x SD polen --> cross of individuals of the same species

SDSC = SD ovule x SC polen  -->cross of individuals of different species --> interspecific hybrids

SDZJ is not a hybrid, so this statement needs to be rewritten.

In the same way:

SCZJ = SC ovule x SC polen --> cross of individuals of the same species

SCSD = SC ovule x SD polen  -->cross of individuals of different species --> interspecific hybrids

SDZJ is not a hybrid, so this statement needs to be rewritten.

Please rewrite the sentence.

Please use another verb instead of conferred, perhaps crossed.

The corrections must be done in all the manuscript on the hybrid definition. Review "hybrid" terminology throughout the manuscript.

Figure 1 needs a better resolution and size.

There is no fibrous layer in the anther. The layer in question is the endothecium. It must be corrected in the entire manuscript.

All abbreviations are not explained in the legend of Fig. 4.

Fig. 3I has the luminosity saturated and it is necessary to decrease the light intensity.

Line 266: pollen of P. Aphrodite à The specific epithet must be in lower case, and the genus was not cited before, so you must write: Phalaenopsis aphrodite.

Some abbreviations in the text and captions of the figures are not explained at the first citation.

The anther development stages must be specified in the methodology. It is necessary for the reproducibility of the work. Can you give more details about previous development stages? Perhaps, you could say the anther sizes.

Author Contributions: The text is literally the same as the instructions for authos of the Int. J. Mol. Sci.! It must be adapted to the present manuscript.

Some suggestions are pointed out in the manuscript.

Author Response

Reviewer 1

Response:

We really appreciate your valuable suggestions to improve the quality of our MS. We have made modifications to the manuscript and we present point-to-point responses below:

  1. Readability needs some kind of improvement, with an English review.

See: For a long time, our group finds that S. miltiorrhiza distributed in China (Sichuan province, SC) does not produce seeds under natural conditions [23].

The text presents an abbreviation for a region: China (Sichuan province, SC). Nonetheless, this abbreviation is used in the present work for the plants, and not for a region. So, I recommend an unequivocal explanation of the SC abbreviation: Salvia miltiorrhiza is distributed in China, Sichuan province (here called SC plants). For a long time, our group finds that the individuals of the SC population do not produce seeds under natural conditions [23].

Response: We have improved the English of the manuscript, please review it again. To reduce this ambiguity, “S. miltiorrhiza distributed in China (Sichuan province, SC)” was changed to “S. miltiorrhiza distributed in Sichuan province of China (here called SC)”. Please check lines 12-13 and 44.

  1. Several symbols and abbreviations are not explained in the figures and text. Graphs should be self-explanatory. Legends must contain all abbreviations and symbols used in each graph.

Response: Modifications have been made as requested by the reviewer.

  1. Both abbreviations SC and SD must be in the same order throughout the manuscript. Please use only one pattern in all text and captions for figures and tables.

SC first and SD after or SD first and SC after

It will improve readability. A revision of this sequence must be done in all manuscript.

Response: To improve readability, “SC and SD (S. miltiorrhiza distributed in Shandong province of China)” was changed to “SD (S. miltiorrhiza distributed in Shandong province of China) and SC”. Please check line 62. Moreover, we applied “SD first and SC after” in the all text.

A revision of this has been done in the all manuscript.

  1. Several species are without italics and their taxonomic authorities are missing. Taxonomic authority is required for the first citation of a species. The species first citation is with the genus in full; after that, the genus is abbreviated, unless there is a risk of misunderstanding. You need to review the entire manuscript in view of these aspects.

Response: We have checked the all text, and modifications have been made as requested by the reviewer.

  1. There are no citations and references for some methodological procedures, such as pollen fertility, SEM and TEM analyses, and histological exam. Which is the literature used in these procedures? The methodology on pollen grain vitality is lacking. Please report this methodology or point to the literature on which this procedure is based.

Response: We have added specific operating steps and references in the corresponding positions of the MS. Please check 4.3, 4.4, 4.6. The references for each section were as follows: pollen fertility (https://doi.org/10.1016/S1673-8527(08)60024-7), SEM (https://doi.org/10.1002/jemt.23448), TEM (https://doi.org/10.1007/978-1-62703-776-1_4).

  1. Review "hybrid" terminology throughout the manuscript.

SDZJ = SD ovule x SD polen --> cross of individuals of the same species

SDSC = SD ovule x SC polen  -->cross of individuals of different species --> interspecific hybrids

SDZJ is not a hybrid, so this statement needs to be rewritten.

In the same way:

SCZJ = SC ovule x SC polen --> cross of individuals of the same species

SCSD = SC ovule x SD polen  -->cross of individuals of different species --> interspecific hybrids

SDZJ is not a hybrid, so this statement needs to be rewritten.

Please rewrite the sentence.

Please use another verb instead of conferred, perhaps crossed.

The corrections must be done in all the manuscript on the hybrid definition. Review "hybrid" terminology throughout the manuscript.

Response: “Artificial hybridization” was replaced with “artificial pollination” in all text. Moreover, we replaced “hybrid offspring” with “offspring” in the caption of Fig. 1G. “Hybridization” was deleted.

The entire manuscript has been revised.

  1. Figure 1 needs a better resolution and size.

Response: We have made adjustments to the resolution and size of Fig. 1.

  1. There is no fibrous layer in the anther. The layer in question is the endothecium. It must be corrected in the entire manuscript.

Response: “FL, fibrous layer” has been replaced with “EN, endothecium”, and we corrected Fig 2 and Fig 3.

  1. All abbreviations are not explained in the legend of Fig. 4.

Response: We added “Ba, bacula; Ca, callose; CW, cell wall; En, Endothecium; Ex, exine; ML, middle layer; MP, microspore; N, nucleus; Ne, nexine; T, tapetum; Tds, tetrads; Ub, Ubisch body; Te, tectum; SG, starch granule.” in the Fig. 4 caption. Please check lines 166-168.

  1. 3I has the luminosity saturated and it is necessary to decrease the light intensity.

Response: We decrease the light intensity of Fig. 3I.

  1. Line 266: pollen of aphrodite The specific epithet must be in lower case, and the genus was not cited before, so you must write: Phalaenopsis aphrodite.

Response: Modifications have been made as requested by the reviewer. Please check line 229.

  1. Some abbreviations in the text and captions of the figures are not explained at the first citation.

Response: We have rechecked the all text, and explanations of abbreviations were added in the text and captions of the figures at the first citation

  1. The anther development stages must be specified in the methodology. It is necessary for the reproducibility of the work. Can you give more details about previous development stages? Perhaps, you could say the anther sizes.

Response: We added “The sizes of anthers at different stages were as follows: pollen mother cell stage (0.05-0.08 mm), meiosis stage (0.10-0.15 mm), tetrad period stage (0.20-0.22 mm), young microspore stage (0.25-0.33 mm), and mature pollen stage (0.40-0.50 mm).” in 4.5. Paraffin section observation. Please check lines 276-278.

  1. Author Contributions: The text is literally the same as the instructions of the Int. J. Mol. Sci.! It must be adapted to the present manuscript.

Response: We apologized for our negligence. “For research articles with several authors, a short paragraph specifying their individual contributions must be provided. The following statements should be used” was deleted. Please check lines 317-319.

  1. Some suggestions are pointed out in the manuscript

Response: We have made modifications according to the annotations in the manuscript, and the specific modifications are as follows:

  • It is confuse. Suggestion: Salvia miltiorrhiza is distributed in Sichuan province. Under natural conditions, it does not bear seeds and its sterility mechanism is still unclear. Through artificial hybridization, there were defective pistil and partial pollen abortion in these plants.

Response: “Under natural conditions … partial pollen abortion in SC.” was changed to “S. miltiorrhiza is distributed in Sichuan province of China (here called SC). Under natural conditions, it does not bear seeds and its sterility mechanism is still unclear. Through artificial cross, there were defective pistil and partial pollen abortion in these plants.” Please check lines 12-15.

  • In the middle of the phrase, the genus can be abbreviated, when it was been cited before.

...affected the fertility of S. miltiorrhiza.

Response: “Salvia miltiorrhiza” was abbreviated as “S. miltiorrhiza”. Please check line 19.

  • The sterility occurs in the plant, and not in the geographical region.

Response: “SC” was deleted. Please check line 21.

  • male-sterile plants or male sterility.

Response: In keywords, “male sterile” was changed to “male sterility”. Please check line 23.

  • It is confuse. Suggestion: The microspore mother cell is diploid and has two successive meiotic divisions, resulting in the microspore tetrad. Each microspore has a thick callose wall and a thin pectin wall...

Response: “Diploid microspore cells are formed by microspore mother cells by meiosis, and the diploid microspore cells subsequently form microspores by second meiosis” was changed to “The microspore mother cell is diploid and has two successive meiotic divisions, resulting in the microspore tetrad. Each microspore has a thick callose wall and a thin pectin wall”. Please check lines 30-32.

  • Italics about “Arabidopsis”

Response: “Arabidopsis” in the all text was changed to italics.

  • See: For a long time, our group finds that miltiorrhiza distributed in China (Sichuan province, SC) does not produce seeds under natural conditions [23]. The text presents an abbreviation for a region: China (Sichuan province, SC). Nonetheless, this abbreviation is used in the present work for the plants, and not for a region. So, I recommend an unequivocal explanation of the SC abbreviation:

Salvia miltiorrhiza is distributed in China, Sichuan province (here called SC plants). For a long time, our group finds that the individuals of the SC population do not produce seeds under natural conditions [23].

Response: To reduce this ambiguity, “S. miltiorrhiza distributed in China (Sichuan province, SC)” was changed to “S. miltiorrhiza is distributed in Sichuan province of China (here called SC). For a long time, our group finds that the individuals of the SC population do not produce seeds under natural conditions [23]”. Please check lines 44-45.

  • Thus, the analysis of miltiorrhiza sterility origin will help people to cultivate sterile male lines,

Response: “analyzing the sterile reason of S. miltiorrhiza will help people to cultivate male sterile lines” was changed to “the analysis of S. miltiorrhiza sterility origin will help people to cultivate male sterile lines”. Please check lines 47-48.

  • genes about the male sterility.

Response: “male sterile” was changed to “male sterility”. Please check line 58.

  • italics about miltiorrhiza.

Response: “S. miltiorrhiza” in the all text was changed to italics.

  •  

Response: Starting with “To further understand the cause of infertility in SC”, we started a new paragraph. Please check line 67.

  • The methodology on pollen grain vitality is lacking. Please report this methodology or point to the literature on which this procedure is based.

Response: The methodology and the reference (https://doi.org/10.1016/S1673-8527(08)60024-7) on pollen grain vitality were added in 4.3. Please check line 266.

  • The idea here is unclear. Rewrite.

Response: “magnification observation” was changed to “further observation of the differences in pollen wall”. Please check line 85.

  • Tables and figures should be self-explanatory without the rest of the text. However, SD and SC are not explained here.

Response: We have added explanations (“SD and SC referred to S. miltiorrhiza distributed in Shandong province and Sichuan province of China, respectively”) for SD and SC in the captions. Please check lines 94-95, 104-105, 158-159, and 165-166.

  • The graphics are confusing. What is the SC versus SD in Fig. 1E-F?

Response: As shown in the caption of Figure 1E-F, they represented the setting rates of SD and SC under natural and artificial bagging conditions, respectively. Please check lines 91-93.

  • SDZJ and SCZJ are not hybrids.

Response: “Hybrid” was deleted. Please check lines 95-97.

  • Which periods were selected to study in the present work?

Response: We introduced the selected period in the following text in part 4.5. Please check lines 276-278.

  • Ubisch

Response: We have made the necessary modifications as required, please check lines 121-123 and 126.

  • (A and B) (C and D) (E to F) (G and H) (I and J)-(A-B) (C-D) (E to F) (G-H)(I-J)

Response: We have made the necessary modifications as required, please check lines 156 and 1577.

  • The specific epithet must be in lower case, and the genus was not cited before, so you must write: Phalaenopsis aphrodite.

Response: “P. aphrodite” was changed to “Phalaenopsis aphrodite”. Please check line 229.

  • Line 274

Response: “As is known to all” was deleted. Please check line 235.

  • The abbreviation IAA and ABA must be explained.

Response: “IAA” and “ABA” were marked full names in their respective positions. Please check lines 237 and 242.

  • This phrase is without subject. photos were taken.

Response: “Observe and count at least 30 visual fields respectively under Nikon ECLIPSE CI microscope for sterility rate statistics, and then take photos.” was changed to “We observed and counted at least 30 visual fields respectively under Nikon ECLIPSE CI microscope (×200) for sterility rate statistics, and then took photos.” Please check lines 266-267.

  • What were selected?

    the appropriate position and multiple under the microscope

Rewrite please.

Response: When we conducted SEM analysis, we believed that it is impossible to accurately describe such an appropriate position under the electron microscope. Nevertheless, we added additional explanations. Please check lines 272-273.

  • Can you give more details about previous development stages? Perhaps, you could say the anther sizes.

Response: We added “The sizes of anthers at different stages were as follows: pollen mother cell stage (0.05-0.08 mm), meiosis stage (0.10-0.15 mm), tetrad period stage (0.20-0.22 mm), young microspore stage (0.25-0.33 mm), mature pollen stage (0.40-0.50 mm)” to describe the different stages of anther development. Please check lines 276-278.

  • “Paraffin sections were created as in previous research[25]” should be modified.

Response: “fixative” was deleted. Please check line 277. Moreover, “Paraffin section was created as in previous research” was changed to “Anther sections were obtained by paraffin inclusion according to previous research”. Please check lines 278-279. Meanwhile, we added standards for selecting anther sizes at different stages. Please check lines 277-278.

  • It is necessary to explain this formula.

Response: We added a reference (https://doi.org/10.1006/meth.2001.1262) to explain the method. Please check line 303.

  • “analysis of” redundancy

Response: We have removed duplicate words. Please check line 306.

Reviewer 2 Report

The authors' work deals with a very interesting subject: male sterility. Various mechanisms have long been known to be involved in this process and it occurs as a result of early degradation of the tapetum, leaving microspore tetrads without nutrition and dying. Hormonal signalling, in particular ABA (https://doi.org/10.1007/s00709-017-1185-x), is known to be primarily involved in this process. And yes, tapetum degradation is thought to be programmed cell death. The authors claim this on the basis of electron micrographs, but this is not enough to say that it is PCD. In this case, there are specific reactions with the cell's DNA that unequivocally prove this. Therefore, either TUNEL needs to be done or caspase-like activity needs to be shown. Either the authors only have to make assumptions about PCC based on their data.
Overall, the authors' article is well done methodologically and the data are very interesting. However, with such experimental material, the authors' conclusions are not fully justified. Either more research needs to be done or the wording about PCD needs to be relaxed.

Author Response

Reviewer 2

The authors' work deals with a very interesting subject: male sterility. Various mechanisms have long been known to be involved in this process and it occurs as a result of early degradation of the tapetum, leaving microspore tetrads without nutrition and dying. Hormonal signaling, in particular ABA (https://doi.org/10.1007/s00709-017-1185-x), is known to be primarily involved in this process. And yes, tapetum degradation is thought to be programmed cell death. The authors claim this on the basis of electron micrographs, but this is not enough to say that it is PCD. In this case, there are specific reactions with the cell's DNA that unequivocally prove this. Therefore, either TUNEL needs to be done or caspase-like activity needs to be shown. Either the authors only have to make assumptions about PCC based on their data.
Overall, the authors' article is well done methodologically and the data are very interesting. However, with such experimental material, the authors' conclusions are not fully justified. Either more research needs to be done or the wording about PCD needs to be relaxed.

Response:

Thank you for your affirmation of our work. Due to the lack of TUNEL and caspase-like activity experiments, we also agree with your viewpoint that using programmed cell death (PCD) in this article is inappropriate. According to the results of paraffin sectioning, “delayed degradation of the tapetum” seems to be more appropriate. Thus, we have revised the conclusion of the MS.

Round 2

Reviewer 1 Report

Some things still need attention.

4. Several species are without italics and their taxonomic authorities are missing. Taxonomic authority is required for the first citation of a species. The species first citation is with the genus in full; after that, the genus is abbreviated, unless there is a risk of misunderstanding. You need to review the entire manuscript in view of these aspects.

Response: We have checked the all text, and modifications have been made as requested by the reviewer.

è Some genera and species are still without italics. They are pointed out in the manuscript. 

10. Fig. 3I has the luminosity saturated and it is necessary to decrease the light intensity.

Response: We decrease the light intensity of Fig. 3I.

è The Fig. 3I has not been corrected. The image remains as before.

Author Response

We would like to thank you again for your detailed revision comments, which are very helpful for improving the quality of our manuscript. Regarding these comments, we present point-to-point responses below. Finally, we hope the manuscript meets your expectations and requirements.

1.Some genera and species are still without italics. They are pointed out in the manuscript.

Salvia in italics and without abbreviation at the beginning of the sentence.

Response: “S. miltiorrhiza” was changed to “Salvia miltiorrhiza” at the beginning of the sentence. Please check lines 12 and 44.

We have rechecked the entire text and confirmed that all species have been italicized. Please check lines 73, 75 and 165

“by paraffin sections” was deleted. Please check line 109.

“ubisch” was changed to “Ubisch”. Please check line 167.

  1. The Fig. 3I has not been corrected. The image remains as before.

Response: We have already retaken and adjusted Figure 3I. Please check line 154.

Reviewer 2 Report

The authors have significantly revised the manuscript. I have no further complaints about the content of the study. The only minor comment to the design of the figures, it would be desirable to move the dimensionality of Bar everywhere in the caption to the figuure, and on the figures themselves to leave only the line. This, however, does not affect the quality of the article in the long run and should be left as a suggestion for future manuscripts if the editorial staff thinks it should be left in this one as well. I believe the manuscript can be recommended for publication in the journal.

Author Response

The authors have significantly revised the manuscript. I have no further complaints about the content of the study. The only minor comment to the design of the figures, it would be desirable to move the dimensionality of Bar everywhere in the caption to the figure, and on the figures themselves to leave only the line. This, however, does not affect the quality of the article in the long run and should be left as a suggestion for future manuscripts if the editorial staff thinks it should be left in this one as well. I believe the manuscript can be recommended for publication in the journal.

Response: Thank you for your recognition of our work, and we have carefully considered your opinion. Thus, we have adjusted the Bar in Fig. 3 and Fig. 4 to ensure that they leave only the line, and we explain the length represented by Bar in the corresponding annotations. Please check lines 154 and 161.